# Towards the Digital Twin (DT) of Narrow-Band Internet of Things (NBIoT) Wireless Communication in Industrial Indoor Environment

**DOI:** 10.3390/s22239039

**Published:** 2022-11-22

**Authors:** Muhammad Dangana, Shuja Ansari, Syed Muhammad Asad, Sajjad Hussain, Muhammad Ali Imran

**Affiliations:** 1James Watt School of Engineering, University of Glasgow, Glasgow G12 8QQ, UK; 2Transport for London (TfL) Head-Quarter, 14 Pier Walk, Greenwich Peninsula, London SE10 0ES, UK

**Keywords:** NB-IoT, digital twin (DT), Industry 4.0, wireless communications, machine learning, 5G, 3GPP, low-power wide-area network (LPWAN), throughput, signal to noise (SNR)

## Abstract

A study of the behavior of NB-IoT wireless communication in an industrial indoor environment was conducted in this paper. With Wireless Insite software, a scenario in the industrial sector was simulated and modeled. Our research examined how this scenario or environment affected the communication parameters of NB-IoT’s physical layer. In this context, throughput levels among terminals as well as between terminals and transceiver towers, the power received at signal destination points, signal-to-noise ratios (SNRs) in the environment, and distances between terminals and transceivers are considered. These simulated results are also compared with the calculated or theoretical values of these parameters. The results show the effect of the industrial setting on wireless communication. The differences between the theoretical and simulated values are also established.

## 1. Introduction

Prior to the emergence of the NB-IoT, the direction and prospects of the Internet of Things (IoT) in terms of its numerous applications were becoming acceptable to the industrial world [1]. These prospects have led to the awareness of advancements in the application of IoT in industries. To this end, NB-IoT standardization for application in industrial environments has gained traction as it meets some of the requirements of Industry 4.0. NB-IoT is among the family of LPWAN technologies which are characterized by low-power consumption, wide-coverage, low cost of manufacturing and deployment, and low data rates—which are some of the major points promoted by the Third Generation Partnership Project (3GPP). In the case of NB-IoT, its service life objective is specified by 3GPP for a terminal battery life cycle of 10 years. The major techniques behind this achievement are the power-saving mode (PSM) and extended discontinues reception (eDRx) mechanisms which are used to improve battery efficiency [2]. This is made possible through the combination of reduced peak-to-average ratio (PA), improved PA efficiency, reduced times for periodic measurements, and promoted a single phased process of objective achievement. However, in the practical application of NB-IoT, its battery life for service is related to the coverage area needs of the terminal and service model. This low-power consumption is further strengthened by its adoption of simple modulation, demodulation, and encoding techniques which also reduce the requirements for the memory space and processor [3]. The market share for NB-IoT is expected at USD 60 billion, but there is still plenty of room for other types of IoT technologies in the market; nevertheless, NB-IoT has a bright future ahead. Under the conditions of current market dynamics, the average cost of the NB-IoT module is below USD 5. In the future, this might decrease based on the expansion of the world market scale. The precise cost and time will, however, be determined by its demand and advancement in technology [4].

The number of connections of terminals and its deployment scalability is an important motivator for the application of IoT on a large scale. NB-IoT has a preliminary objective of 50,000 connections per unit cell, which has been met based on its preliminary computational evaluation; therefore, NB-IoT has satisfied this objective. Meanwhile, certain factors such as the NB-IoT terminal service model in the cell affect the realization of this designed objective in practice. To apply this objective in industrial indoor environments, further evaluation is required. NB-IoT does not support mobility, which is because cell handover and redirection are not possible in the connected state, but cell re-selection is only allowed or possible in an idle state (e.g., NB-IoT R13) [5]. However, one of the requirements for vertical industrial needs is the mobility management of NB-IoT in the connected state. The ability of NB-IoT leverage on long-term-evolution (LTE) platform, to support reselection in a connected state will be an area in need of optimization. In the decision-making process of the industrial implementation of wireless communication technology, the cost of network deployment, complexity, and scalability are major constraints that must be dealt with [6]. Therefore, to deploy an NB-IoT wireless network in a non-existing LTE, frequency division duplexing (FDD) industrial environment, the deployment would be closer to the deployment of a brand-new network. This is because it involves the building of a new infrastructural wireless network with the core network (either software- or hardware-based) and gradual adjustments to the structures of transmission. This implies that the cost of implementation will be higher for a site with no ready-made idle spectrum because the spectrum adjustment will be a standalone mode of deployment if a GSM network is used. On the other hand, the deployment of the NB-IoT network in sites that already have LTE FDD will provide great relief for its actualization as the existing equipment and spectrum will be put to use and this will make it simpler to implement. In some cases, however, regardless of whether the LTE infrastructure exists, some situations require upgrades for the equipment or core network. For the core network, a software upgrade would be required to support NB-IoT as well as new radios or filters.

The maximum data transmission rate for NB-IoT is less than 300 kbps, a rate which is not suitable for some applications but enough for information-processing in some industrial environments [7].

In this paper, therefore, the purpose is to examine the impacts of wireless communication of NB-IoT in industrial buildings; the material properties of the industrial structure, the interior elements (such as machines), and the noise generated within the industrial indoor environment based on measurements from its physical layer wireless parameters.

After the introduction of this paper in Section 1, other research works carried out in previous years were highlighted in Section 2. The technology synopsis in Section 3 summarizes the application of wireless communication in industrial indoor setups, NB-IoT configuration limits, and the amount of throughput achieved by NB-IoT terminals. Section 4 is the methodology used in this research work as it explains the experimental setup, simulation scenario, and data collection technique. The experimental results obtained are discussed in Section 5 while Section 6 highlights some of the challenges and future aspects of the research, and finally, Section 7 concludes the paper.

## 2. Related Works

NB-IoT is a type of IoT that has many applications in the wider world and this includes both outdoor and indoor applications. However, its applications in an industrial indoor environment are still on the rise, which is due to the requirements needed to support and sustain industrial networks. The evaluation of its capacities in indoor environments was carried out, and these studies reveal the performance of NB-IoT in the presence of indoor factors on this technology. The authors in [8] used NB-IoT to reduce the signaling overhead of early data transmission (EDT) and pre-configured uplink resources (PURs) for industrial IoT systems. They presented the architectural framework as well as the procedures required for its management. The use of NB-IoT was implemented in an intelligent fire protection system [9]. The authors highlighted the challenge of smart smoking. The challenge of the coexistence between regular and stochastic traffic in NB-IoT systems has been addressed in [10], wherein the authors addressed this by building an analytical model which captured the importance of the random-access procedures for two traffic types of interest. This was implemented over an application query of intervals and was parameterized using some sets of practical measurements. The authors further optimized the mean delay behavior of regular traffic. However, the authors did not implement these findings in the industrial indoor environment. The application of NB-IoT in the field of agriculture in a greenhouse setting was implemented in [5]. The study confirmed the low-power consumption rate of the system, its real-time transmission, and remote monitoring capability. Nevertheless, the structural and environmental effect of the greenhouse on the NB-IoT transmission was not studied. The authors in [11] used empirical measurements to evaluate the connectivity and coverage range of the NB-IoT network in outdoor, indoor, and underground environments. The study showed that NB-IoT performed better in both outdoor and indoor environments than underground. In the indoor measurement, the study did not elaborate on the physical composition of the indoor environment and did not state whether it is an industry. The authors in [12] discussed and confirmed the growing adoption of industrial-scale NB-IoT networks, they discussed the telecommunication operator’s findings and considerations during measurements, tests, and implementations, and compared the obtained results with the 3GPP NB-IoT ratified technology. Some of the challenges facing NB-IoT in the market were also highlighted, however, the findings the from tests and measurements performed did not detail the effects of the surroundings wherein the network was implemented. The NB-IoT network was deployed by [13] in an indoor environment, and in this study the authors evaluated the performance of the network and found that the technology met expectations, however, they also highlighted the limitations associated with it in terms of packet delivery sizes and delay. The paper made these evaluations in an indoor building but did not explain the impact of the material properties on the network. Using an experimental measurement technique, reference [14] measured NB-IoT signal attenuation in a deep-indoor environment and compared signal propagation between the underground and above-ground scenarios. The results obtained show that NB-IoT underground measurements do not conform to the theoretical model as more attenuation is experienced.

For smart cities, the authors in [15] proposed DT as a fundamental application through energy Internet construction or integrated regional energy systems. They detailed the benefits of DT, including its economic advantages, based on this proposal and their introduction of an energy-planning platform, CloudIEPS. In order to demonstrate the application of industrial IoT, Pavatar, a system for monitoring and diagnosing ultra-high-voltage converter stations (UHVCSs), was used as a case study by [16]. To free the workforce and provide credible decision support for industrial operations, the authors envisioned how DT could automatically monitor and comprehensively simulate the factory throughout its entire lifecycle. This includes production to manufacturing, operations to maintenance. In [17], the authors presented the evolution levels of DT, implementation layers, and future aspects of DT. The implementation layers were supported with technological elements as an introduction. These, the authors suggested, could serve as a step-by-step implementation method that could be applied to digital twin realization. The exploration for the implementation of DT in additive manufacturing (AM) in product design was discussed in [18]. The authors proposed the product conceptualization, design, and virtual verification of a biomedical scaffold in an AM setting using DT technologies. They also presented the potential and limitations of the use of DT. The paper did not present technical proof regarding the practical application of DT in the proposed biomedical scaffold.

## 3. Technology Synopsis

The operation of wireless technologies in an industrial indoor environment is guided by the standards that established them. However, the unfavorable condition of the industrial environment poised to these technologies allows for the possible fine-tuning of these standards to their extreme boundaries. This section therefore explains how industrial indoor settings generally affect wireless communications with an interest in NB-IoT terminals by focusing on wireless communication operation in an industrial setup and by taking a closer look at industrial network designs and topologies as different types of industrial traffics and metrics measuring network performances. The technology configuration boundaries, as well as its achievable throughput, are also discussed in this paper.

### 3.1. Wireless Communication in Indoor Industrial Settings

The indoor industrial environment is a characterization of tough electromagnetic interference, critical temperature levels, humidity, and different stages of mechanical stress. This is coupled with the narrow requirements of low latency and delay for real-time scenarios, reliability, and the determinism of industrial systems. Upon these factors, wireless communication is expected to function to meet up with these requirements and survive the characterizations [19]. Industrial settings can be classified into various forms; however, process automation and factory automation are considered in this paper. Moreover, a single industrial setting could involve these two kinds of setups, and an example is a chemical production industry. Therefore, wireless communications in industrial environments are set up based on three major factors that meet up with the type of classification of, e.g., network topologies and designs, types of network traffic, and network performance indicators.

#### 3.1.1. Network Topologies and Designs

Figure 1 shows an advanced structure of a factory automation system with four major levels. These include the field, automation or control, coordinator, and management levels. Field levels accommodate several wireless sensing nodes or wireless Industrial Internet of Things (IIoT). These nodes are attached to devices including robotic arms that carry out discrete actions which require precision in production such as in an electronic processing plant. The automation level connects to the management level where decisions are made to control or adjust the operation of the field levels [20,21].

The infrastructure mode topology is more common in wireless industrial environments, mostly because of the complexity of the system in terms of the required reliability level, security, and real-time needs. The Independent Basic Service Set (IBSS) or ad hoc network topology are also implemented in some industrial wireless communication for some specific purposes but are mainly on a small scale due to the limitation associated with it. Among these limitations are security and privacy, since it is prone to attacks such as man-in-the middle; energy efficiency; and the multicast transmission activity of the network. Wireless mesh network topology also exists in industrial environments in complex set-up systems [22]. In this paper, an infrastructure star–tree collaborative topology mode was used to improve the wireless communication system in an industrial indoor scenario environment.

Industries including the glass, mineral treating, and chemical industries which require continuous industrial processes adopt the process automation system that requires many layers of levels for discrete actions. This implies the involvement of many industrial nodes for successful operations [23]. Figure 1 shows structured process automation where the lower level comprises nodes that sense, gather data, and transmit some to concentrators which further interact with the management level through the coordinator level.

#### 3.1.2. Types of Industrial Network Traffic

The authors in [24,25] classified industrial traffic into cyclic and acyclic traffic which are likely to originate from various sources or sections in an industrial setup with different time requirements. Examples of cyclic traffics are the periodic exchange of sensing data such as measurements and set points. These data have a small payload size but are in most cases time-specific. Acyclic traffic are generated by unpredictable events such as process alarms that require immediate action. The data transferred by this type of traffic are also small in size [26]. Both types of traffic have different levels of criticality assigned to them. Regardless of the type of traffic, the transfer of data implies the transfer of frames with the possibilities of higher payloads which directly influences the throughput and bit error rate.

In this paper, an industrial indoor scenario is modeled and simulated, a scenario that comprises some features similar to those found in a chemical processing industry. The presence of heavy-duty machines or equipment in chemical industries makes it one of the preferred scenarios for this study. Therefore, the following traffic can be identified and classified in this scenario, as found in [27]. The structural overview of this industrial scenario is shown in Figure 2, which is designed into sections that have separate or unique characteristics. This implies that the level of data rate, latency, and reliability requirements differ from section to section in the application of NB-IoT. The following sections make up the factory, and the type of data they gather and transmit are also highlighted [28].

Space for production: Finished goods are produced in the factory in this space through the processing of the raw materials. As part of the production process, raw materials are prepared, mixed, pre-processed, and post-processed, quality assurance is undertaken, product certification is obtained, packaging units are constructed, and assembly lines are installed. This section describes how NB-IoT applications gather and transmit four types of data. Monitor traffic (MT), non-critical alert traffic (NCAT), critical alert traffic (CAT), and critical control traffic (CCT) are included in this category. It is necessary to maintain medium-to-low latency and low reliability in this section due to the high data transmission rate ranging from seconds to hours.Waste storage: By-products and waste generated during production are stored in this unit, which must be monitored as it involves toxic waste and poisonous gases that can endanger the safety of industrial employees and any occurrence must be immediately reported. It is therefore in this section that emergency traffic (ET) and medical traffic (MT) are transmitted. The system must be reliable and have minimal latency.Storage of finished products: This is the place where finished products are kept. Since NB-IoT technology provides adequate coverage for MT and NCAT traffic, NB-IoT is a well-suited technology for maintaining the count of the products, the state of the products, and the condition of the warehouse.Product distribution point: This unit is connected to the warehouse and assembles the finished products to be distributed. For smooth operation, this unit relies on MT.Coordination/control room: This is the factory’s observation room, which monitors all factory activities to ensure compliance with all standards, rules, and conditions required for smooth operation. It stores and transmits the information collected from various departments to the appropriate departments for action, including management, accounting, safety, and production, and thus collecting MT.Staff offices: This section includes the factory staff which are divided into departments. They receive MT on their handheld devices or dedicated computer systems so that they can act upon it.

#### 3.1.3. Network Performance Metrics

Over time, the implementation of wireless communications in indoor industrial environments has increasingly demanded more requirements to meet the changing challenges in the modern world of Industry 4.0. The most paramount amongst these include timeliness, reliability, flexibility, security, openness, and cost-effectiveness [1].

Many industrial activities have specific time constraints and any wireless network that can meet this requirement among others is highly adopted. Two main characters have been explained by [29] to define timeliness: real-time and determinism. The wireless transmission of a measurement value from an NB-IoT sensor to its base station at a precise instant is an example of determinism. Within a specified deadline, the ability of a station to correctly deliver a message is referred to as real-time. The levels of latency, delay, and jitters in industrial wireless communications account for reliability. The flexibility and scalability of an industrial network to support the ever-increasing needs for reconfiguration, expansion, and reassembly is also a requirement in focus. The need to transmit the acquired data from the industrial processes through the Internet or cloud to management control levels involved the need to secure these data from unauthorized access or interceptions from competitors and hackers. Moreover, security in an industrial wireless network can be applied to any level of the network hierarchy to secure the intended data [29].

The authors of International Electrotechnical Commission—IEC 61784-2 [30] highlighted some important performance indicators for industrial networks. Among them are throughput (real-time), delivery time, and bandwidth (non-real-time). The ability to assess the actual real-time capability of an industrial network is given by the first two indicators. Considering the application layer (real-time) traffic, throughput is the number of transmitted octets per second on the identified link of an industrial network. Using an example of a payload message carrying real-time data, the delivery time is the required time to transmit this message from a source node to the destination node of an industrial network. To evaluate the degree of openness in an industrial network, the bandwidth indicator is more suitable as it is the percentage of the network bandwidth required on a specific link for non-real traffic. An example is that of non-real traffic to the Internet.

The transmission quality of NB-IoT in terms of data and information transmission as compared to other LPWAN technologies such as LoRaWAN and Sigfox is inherent in the maximum or gross data rate and link budget or maximum path loss for its uplink and downlink channels. NB-IoT has a 27 kbit/s and 164 dB for its uplink data rate and downlink path loss respectively. In comparison, these values are about 80% higher to LoRaWAN’s gross data rate and 9% more in value to its path loss, while for Sigfox, these represent about 98% and 3.5% higher for uplink data rate and path loss respectively. However, the length of the maximum payload in terms of data per message for NB-IoT is >1Byte while Sigfox is 12Byte and LoRaWAN has 51Byte and 11Bytes for EU and US laws respectively. Further to these, assessing the quality of transmission among these technologies, throughput, reliability, coverage, and data rate are most paramount parameters to be considered. For throughput; the data volumes per day or the total number of messages per day provided by NB-IoT for its uploads or downloads channels are much higher than those of both LoRaWAN and Sigfox. This is due to the fact that, unlike the former, the latter are constrained by the legal duty cycle that applies to the usage of industrial, scientific, and medical (ISM) band. The duty cycle is the maximum allowable percentage of time for which an end device may occupy an ISM band channel. In terms of reliability, the use of licensed LTE frequency by NB-IoT prevents signal interference from other devices as compared to both LoRaWAN and Sigfox which are always faced with the challenge of potential interference due to the use of ISM frequency bands as other devices contend to use the same frequency. NB-IoT is also enabled by a mechanism that permits the re-transmission of signals, which often increases its reliability in transmission. The re-transmission of LoRaWAN and Sigfox is restricted by country laws as they operate on a government-governed ISM frequency band. Transmission reliability is also in favor of NB-IoT when it comes to the available amount of power that can be used for transmission as it increases penetration through walls and a 23 dBm is allowed; however, LoRaWAN and Sigfox are limited to 14 dBm by EU laws [31].

### 3.2. NB-IoT Configuration Boundaries and Topology in an Indoor Industries

The increased adoption of NB-IoT in industries is a result of its capabilities to meet up with the requirements and challenges of industrial setups. These capabilities are well stated by 3GPP standards from release 10 through to release 15. Amongst NB-IoT capabilities suitable for industrial usage is its extended coverage area applicable to indoor environments, low power consumption rate for longer battery life, low-cost devices, scalability in changing network conditions, and security [1,32].

NB-IoT capability to cover a wide area is due to its transmission tone and power consumption mechanisms. The repetition of messages for a failed transmission between the NB-IoT device and its receiver over an area allows for reliability in transmission in its low-power and wide-area connectivity. However, this success is owed to three power classes (3, 5, and 6) introduced by 3GPP in releases 13 and 14. Power class 3 allows for the use of the transmission of power up to 23 dBm; class 5 is the transmission power of 20 dBm; and class 6 is 14 dBm. Since transmission power is directly related to coverage range, then using a higher power class reduces the coverage range. This implies that when a data transmission fails, the repetition mechanism takes over and as long as this mechanism continues to try and meet up with its reliability task, the channel becomes congested. Therefore, it is important that one of the trade-offs necessary to be made in the deployment of NB-IoT in industrial setup is between battery life and the expected coverage range. One of the impacts of this is that any amount of energy savings made through a low-power transmission are either lost or exceeded by the repetition of the signal that is needed to deliver a single message.

Depending on the release type (R13–R15) features applicable to an NB-IoT module, the following configurations regarding its transmission features can be applied [32].

Bandwidth selection: For a 200 kHz bandwidth, the standalone deployment mode is required and for 180 kHz bandwidth, in-band or guardband will suffice.Paging and random access on the non-anchor carrier, release assistance indication, NPRACH range enhancement.Connected mode mobility, wake-up signals, multicast transmission, and group messaging.Early data transmission (EDT), improved access control, small-cell control, cell re-selection.Coverage extension.Power saving mode (PSM), extended discontinues reception (eDRX), power class, new category NB2, battery efficiency security for low throughput (BEST).

Since NB-IoT leverage long-term evolution (LTE) technology, its network topology and architecture are similar to the latter. However, based on the 3GPP specifications, some distinctions are made for the easy transmission of NB-IoT data across the LTE network. Figure 3 shows a network architecture of NB-IoT, the user equipment (UE) is an NB-IoT module fitted with an SIM card from a telecommunication operator providing NB-IoT services. The UE communicates to the Evolve NodeB (eNodeB) of the operator, a distance away through a wireless interface. Using the wireless interface S1, the eNB transmits data to the NB-IoT network core called the evolved packet core (EPC), which in turn forwards the necessary packets to the application server handling NB-IoT data. This transmission is carried out using the SGi interface.

Both Figure 4 and Figure 5 explain the responsibilities of the individual device regarding the transmission of data at the EPC. The EPC comprises Mobility Management Entity (MME), Serving Gateway (SGW), Packet Data Network Gateway (PGW), Home Subscriber Server (HSS), and Service Capability Exposure Function (SCEF). The function of these is as follows: MME has all to do with the User Equipment (UE). This ranges from the authentication of users, through the management of mobility and network access to the selection of SGW for a UE at the initial joining stage and at the point of intra-LTE handover. SGW provides the needed user plane data services to one or a group of eNBs. For a particular UE, the SGW sets up or disconnects sessions after receiving instructions from MME. It also serve as an interface between MME and PGW, and between eNB and PGW for signaling and Internet Protocol (IP) packet handling. PGW enables UE access to the Internet and acts as an IP router. In Figure 5, eNB is a radio access technology that defines two main optimization levels, the user plane, and control plane optimization traffic. The system selects the best path for the transmission of the user and controls plane traffic for both downlink (DL) and uplink (UL). Interface S1-MME handles all the control plane data while S1-U handles the user plane data, however, some setups can transmit both types of data using the same link or interface, for example, some software-implemented EPC. Using the control plane, non-IP data are transmitted in two ways, either through the SGW using the S11 interface or through SCEF using the T6a interface. When this set of data passes through the SGW, it is further transmitted through PGW using interface S5 before reaching the application server. In contrast to the control plane non-IP data, SGW can transfer both IP and non-IP data from the user plane to PGW and onward to the application layer through interfaces S5 and SGi, respectively [13].

The transfer and collection of both data and voice are carried out at the EPC for the NB-IoT system. Voice is an IP application and its integration with data at the MME enables telecommunication operators to deploy and operate a one-packet network for LTE with NB-IoT applications. Figure 4 shows the transmission of data through the NB-IoT network based on both IP and non-IP options. NB-IoT networks are largely designed to handle a small amount of data suitable for specific industrial data needs; on these bases, non-IP data are suggested to transverse across the network since this reduces the data transmission volume. NB-IoT networks allow the use of UDP transport protocol with IPv4 or IPv6 configurations, however, this also depends on the capability of the radio module. TCP can be used over the air interface for NB-IoT, but HTTP and HTTPS are not allowed to be implemented on TCP due to their overhead [33].

### 3.3. The Throughput of NB-IoT Terminals

Maximum throughput can be measured at all layers of a wireless communication system and the values obtained from each layer are likely to differ. This is because each layer of the system adds extra protocol overhead, which means that a certain amount of extra data are added to the original data transmitted as it moves up the layers. Therefore, the maximum throughput measured at the physical layer will be higher than the measured throughput value of the layer above it. This value will continue to decrease subsequently for the layers above. This is the effective throughput. A wide range of internal and external factors affect effective throughput (also known as goodput) [34]. Among them are:Overheads associated with the link layer and application layer protocols;Flow control and network contention;Congestion on the network and the number of users;Asymmetries between download and upload speeds;Conditions of the network channel;Incompatible hardware and software implementations that prevent optimal throughput.

In addressing throughput using the physical layer properties, Shannon’s capacity becomes an important formula in understanding this phenomenon. In this case, the maximum throughput at the physical layer will be subjected to the amount of bandwidth at this layer which in turn is governed by the analog signal processing definition of bandwidth expressed in Hertz (Hz). The maximum throughput measurable at this layer is therefore a function of or limited by bandwidth as represented in Equations (Equation 1)–(Equation 3) [35,36,37,38].
(1)$$C=W\log_{2}\left(1+\frac{S}{N}\right)$$

In Equation (Equation 1), *W* is the bandwidth measured in Hertz, *S* is the signal power, and *N* is the noise power considered to be additive white Gaussian noise (AWGN). Signal power refers to the received power of the system.
(2)$$C=W\log_{2}\left(1+\frac{h_{a}P_{t}}{WN_{0}}\right)$$

Equation (Equation 2) measures the capacity of the channel with the effect of fading. $N_{0}$ is AWGN, $P_{t}$ is the transmitted power, and $h_{a}$ is the channel attenuation.
(3)$$C=W\log_{2}\left(1+\frac{H^{2}P_{t}}{WN_{0}}\right)$$

Equation (Equation 3) considers the channel gain *H* and transmitted power in the presence of AWG noise $N_{0}$.

## 4. System Model

In this section, the simulation setup is explained, which is motivated by the choices of the industrial structural plan; equipment occupying the industry; and the configuration and placement of NB-IoT terminals. The simulation scenario is further explained and the method used in data collection.

### 4.1. Simulation Setup

To carry out this study, a three-dimensional wireless software capable of predicting radio wave propagation and wireless communication systems called Wireless InSite (WI) was employed. The software uses ray-tracing models and high-fidelity electromagnetic solvers to analyze site-specific scenarios. Among the applications of this software is the prediction of communication channel characteristics that apply to complex indoor, outdoor, or even mixed-path environments [39].

#### 4.1.1. Industrial Structural Components and Plan

The physical structural layout and plan of the modeled industry followed a chemical processing factory that meets the standards described in [40]. The structural components include the floors, walls, ceilings, doors, and windows which are modeled in different sizes and materials. The following steps were deployed in setting up industrial indoor wireless propagation analysis in WI while the assigned properties of the building material are presented in Table 1.

The floor plan: A tiff format image of the floor plan was imported into the software using the Project–Open–Imagetabs. On the image property tab, the required entries are a short description of the image or floor plan, the coordinate for the orientation of the image, in this case, the Cartesian coordinate was chosen, top offset for image adjustment to the grid lines, and the pixel spacing for both x and y dimensions. This spacing allows for the scaling of the image to the appropriate size which is calculated as the ratio of the practical measurement of a portion of the floor plan and the pixel dimension of the image. To verify the correct size of the uploaded image, a measuring tool is used to measure part of the image.Floor plan geometry: On the Project–New–Feature–Floor plan tab, both the base and top heights (m) for the floor plan are specified. The floor plan editing window allows for other adjustable working tools such as the snap-to, grid spacing, and material modification tools. However, the floor cannot be created until the walls are first created as explained below.Barrier walls: The walls are created using the specified material types. These walls are repeatedly created using the New–walls tab.Windows and doors: The windows and doors are created using a similar step as the walls are created. However, values for width (m) and heights from the top and bottom of the windows and doors are required to be specified. Meanwhile, to simulate open doors or windows, a free space material is used, however, in this paper, all doors and windows are simulated in a closed state using the ITU 2.4 GHz wooden material.Ceilings or rooftops: the ceilings are also created using similar steps in the floor plan and following the floor geometry.NB-IoT waveform and antenna pattern: A sinusoidal waveform was used for this simulation through the waveform tab and the carrier frequency was also set at 900 MHz. An isotropic antenna pattern was also selected for wireless transmission and reception.

#### 4.1.2. Industrial Equipment

Industrial equipment comprises the machines needed by the industry to carry out its production or processing activities for its final products. In this paper, the equipment includes machines and other industrial objects that are distributed in different sections of the industry. These include assembly lines with robotic arms, conveyor belts, waste tanks, stackers and storage racks, forklifts, delivery vehicles, office tables and chairs, monitoring screens and systems, and other screens. This equipment is modeled using a third-party software and imported into WI using the Stereolithography (STL) file format. Third-party software are software other than WI that are utilized to create objects, equipment, and buildings or structures. They provide more flexibility and features for such purposes. For these equipment or objects to fit into WI, they are re-scaled and re-positioned in the modeling using the appropriate tools. The material type is attached to this equipment individually after being imported into WI. Most of these equipment are made of metal and metal sheets, and in addition to the walls and barriers that cause obstruction to wireless propagation in this kind of environment, the equipment are obstacles as well, thereby causing a reflection of signal and as a result, causes signal fading.

#### 4.1.3. Configuration and Placement of NB-IoT Terminals

In this paper, we consider the transceiver unit of NB-IoT and as such, the physical layer properties and configurations are of the utmost priority in this industrial indoor scenario. Therefore, configurations that affect the transmission of NB-IoT were made. To start with, a bandwidth of 180 kHz was set and deployed in the in-band mode as specified by 3GPP in TR 45.820 [41]. A class 6 power transmission plan was set for all terminals which implies that the maximum transmissible power will not exceed 14 dBm. The placement of several transmitters and receivers in the environment is performed with the intent of measuring the communication between them. These placements or attachments are to the industrial equipment using the transceiver set point tab. This method allows for the individual transceiver to be set and allows for a short description of each terminal. To attach terminals to the machines, positions, and heights, the layout properties–edit control points–edit vertexprocess is used. These placement of NB-IoT terminals are shown in Figure 6

### 4.2. Simulation of Scenario

To run the simulation of the scenario, firstly, the boundary volume called the study area needs to be defined. This boundary dictates to the WI the specific portion of the project or model area that needs to be captured for the simulation with all the features input. The study area tab is used to achieve this by choosing the fit to feature as the boundary method because it automatically fits the study boundary to contain all the features in the simulation model. On the study area property window, the propagation model is chosen; the desired value for reflections from surfaces is entered; the required number of transmissions through walls, doors, windows, and other barriers are entered; and the number of needed diffraction from all edges and corners are also specified. Specifying these values informs the ray-tracing mechanism of the software of how many physical interactions are expected from the propagation paths with the geometry of the model.

#### Data Collection

Upon selecting the numbers of desired outputs from the Output request tab, the simulation was run until completion by monitoring the progress of the calculation log while it runs for successful completion. Among the requested outputs are complex impulse response, propagation paths, received signal power, time of arrival, path loss, path gain, the direction of arrival, the direction of departure, throughput, bit-error-rate, and signal-to-noise ratio.

The outputs are accessible through the output tab which is also presented in an expanding tree format for collection. Using the expanding tree format, results for inter-connecting terminals can be collected and documented for analysis.

## 5. Simulation Results

To evaluate the performance of the system, the following results were obtained and analyzed. These include the throughput of the physical layer by considering the received power in dBm, the throughput and the distance in m, the effect of SNR on the throughput in industrial environments, the relationship between distance, SNR, and received power. Using the Shannon capacity equation, the throughput in kbits/s was also calculated and the results plotted alongside the simulated results for comparison. In the modeling and simulation, the 3GPP Release 15 specifications guiding the maximum power transmission of NB-IoT was employed. In this release, the allowable maximum power transmission for NB-IoT is no more than 14 dBm. In this modeling, the maximum power transmitted by the NB-IoT antenna did not reach this value. Hence, the received power at the transceiver was recorded with the minimum and maximum received power values as −110 dBm and 6.9232 dBm for uplink transmission, respectively, and −95.9412 dBm and 10.3213 dBm for downlink transmission. The calculated throughput is the upper limit or theoretical maximum value at the physical layer that is attainable by the NB-IoT terminals, as described by Shannon’s theorem. To calculate throughput using Shannon’s theorem, a predictable amount of 3.6 dB noise was introduced. This is necessary because the theorem incorporates noise and errors which it assumes to be fairly consistent and predictable. This consideration is evaluated in the SNR ratio as the received signal power to the noise power. The results obtained also evaluate the effect of distance on the throughput levels. The distance in meters is how precisely the NB-IoT terminals are far apart from the transceiver. The impact of this on throughput is evaluated to understand the error-free data that can be safely transmitted through the channel. The performance of the channel is also evaluated based on the levels of the SNR values. This is to understand how effectively the received power and the noise power impacted the throughput of the system. Further investigation was carried out into the relationships between both SNR and received power on distance.

### 5.1. Throughput Levels Based on Received Power

The simulated and calculated results for the throughput levels is shown in Figure 7, this is for the uplink channels of the NB-IoT terminals. This is also function of the amount of power received by the transceiver antenna. As shown in the figure, both the simulated and calculated throughput of these channels increase as the power received by the transceiver increases. For the uplink channel simulated values, the lowest throughput value is 1.573 kbps at a received power of −106.708 dBm. While the lowest calculated uplink channel throughput value is 2.056 kbps for a received power of −110.227, the highest uplink channel throughput values for both simulated and calculated are 168.784 kbps and 194.890 kbps at the received power levels of 9.785 dBm and 6.9323 dBm, respectively. The throughput level was minimal in values for 1–10 NB-IoT terminals until the received power level improved for the 11th terminal. On the other hand, Figure 8 shows both the simulated and calculated downlink channel throughput levels, where the lowest simulated throughput value is 1.854 kbps at a received power of −93.200 dBm, and the calculated lowest throughput value is recorded as 4.0755 kbps at a received power of −95.9412 dBm. Similarly, the highest recorded downlink channel throughput values for both simulated and calculated values are 179.370 kbps and 205.5787 kbps for received powers of 10.1385 dBm and 10.3213 dBm, respectively. From the results, the downlink channel performed better as the transceiver antenna has more power to transmit through the environment. In comparison, the calculated or theoretical throughput is higher in value compared to the simulated throughput level. This is true as the theoretical values of the throughput are less in practice because of the type of noise that is assumed in Shannon’s theorem, compared to in reality, where other forms of noise exist and are not predicted in their entirety. Similarly, the theoretical throughput is in a bandwidth-limited system operational state. This is where the NB-IoT bandwidth is deployed in an in-band or guard-band mode and fixed to a value of 180 kHz. Therefore, the fluctuation in the received power levels determines the boundaries of the physical layer capacity of the channel with a fixed and predictable noise level.

### 5.2. Reliance of Throughput on NB-IoT Distance from Transceiver

In the model, 25 NB-IoT terminals were strategically attached to the industrial machines for data collection and transmission to the transceiver tower. These distances from the NB-IoT and the transceiver are distinct distances. Among them are some NB-IoT terminals clumped around the waste storage and disposal section; similarly, some NB-IoT terminals are attached to the robotic arms in the production space, as shown in Figure 6, and these positions are among the farthest in the scenario from the transceiver. The maximum distance recorded is 70.584 m away from the transceiver. Figure 9 reveals the relationship between the throughput values and the distance for both the simulated and calculated uplink channels. The highest amount of simulated uplink throughput (168.764 kbps) recorded is against a distance of 10.001 m while the highest calculated uplink throughput channel is at the same distance. The lowest uplink throughput (1.573 kbps) is against a distance of 70.584 m for the simulated result as well as for the calculated results. In a similar account, the highest value for simulated downlink throughput (179.370 kbps) is recorded against the lowest distance of 10.001 m and the same for the calculated values. Figure 10 shows the downlink values for both simulated and calculated results for the throughput against the distance as well.

In this scenario, it implies that the closer a terminal is to the transceiver, the better the throughput value. On the other hand, the farther it is away from the transceiver, the lesser the throughput value. However, at a distance of 27.654 m, an unexpected increase in throughput value was recorded, corresponding to 63.107 kbps and 96.773 kbps for uplink channels for the simulated and theoretical channels, respectively. This is a result of less observed reflectiveness of the industrial machines to the paths of the propagation rays reaching the transceiver. The figure also shows some NB-IoT distance values of 47.24967401, 50.17187857, 52.51184646, 55.10515618, 57.94948749, 60.29264572, 62.96991933, 65.06205596, 67.90667731, and 70.58424096 m having the least throughput values.

### 5.3. The Effect of Signal-to-Noise (SNR) on Throughput

SNR evaluates the influence of noise in a wireless system, especially its effect on received power. Figure 11 indicates that the upper limit of the channel capacity cannot be exceeded regardless of how the SNR ratio is regulated. Most NB-IoT terminals have SNR values that fall between 0 and 200. Focusing on the amount of received power, an increase in SNR ratio effectively corresponds to a dominance of the signal power and as such, relatively throughput values are recorded which also correspond to the theoretical expectation, (Shannon’s theorem).

### 5.4. Evaluating Distance and SNR

As expected and shown in Figure 12, the SNR continues to drop as the distance between the NB-IoT terminals and the transceiver increases. Moreover, this increase is a result of the equipment obstruction to the propagation of wireless rays, (non-line-of-sight). To maintain a high level of SNR as the distance increases, the received signal power needs to be increased and the noise power to be maintained at as a low a level as possible.

### 5.5. Received Power Dependency on Distance

Figure 13 shows how the amount of power received by transceiver depends on its distance from the NB-IoT terminals. From the figure, better power is received when the terminals are closer to the transceiver, however, the reflective nature of the industrial machines also played important roles in these values.

## 6. Challenges and Future Works

There are some challenges associated with the practical implementation of NB-IoT wireless communication in industrial settings since there are a number of variables to consider. In other words, this applies regardless of whether there is a LTE infrastructure in place. In the same way, some of these challenges are also present when simulating the same scenario. One of these challenges is the use of a third-party software in the process of modeling these industrial scenarios and equipment, which makes the integration into its application difficult. WI does not provide or allow the use of propagation models not listed in its software, so choosing a suitable or intended propagation model for simulation was a challenge. Herewith, we also list some challenges that need to be addressed for a more reliable and effective application of NB-IoT in an industrial indoor environment.

Energy-harvesting techniques: Maintaining the low consumption rate of energy as specified by 3GPP (through its PSM and eDRx schemes) by NB-IoT is becoming challenging. This is because of the re-transmission mechanism used to achieve coverage and reliability as the energy saved is depleted in this aspect. Therefore, an energy-harvesting mechanism that incorporates the sourcing of energy from the surrounding environment such as vibration from industrial machines would add more life to this technology.Big data and NB-IoT capacity: The design capacity of NB-IoT to handle and transmit data periodically and the rate of big data generated in the industrial environment coupled with other variables that need to be considered for the implementation of DT needs reconciliation. In this regard, the implementation of ML has become paramount to overcoming this challenge.

Future work will therefore include the improvement of these sets of challenges. In addition, the use of machine learning (ML) in the following aspects is worth mentioning:ML and wireless prediction: The use of ML techniques to predict the wireless communication of NB-IoT terminals in an industrial indoor environment while focusing on the communication parameters and set of environmental conditions gathered in the study. This is aimed to provide a mechanism that would allow for the extrapolation of the study to other industrial facilities that share some common features.ML and Link Adaptation Mechanism: According to some researchers, a hybrid link adaptation strategy can be used to study how coverage enhancement features affect network reliability and latency. In this strategy, latency and coverage were optimized. In order to achieve this, they formulated and solved an optimization problem in which the optimal value of repetitions, bandwidth, and modulation and coding scheme (MCS) was found so that latency is minimized and reliability is maintained. It is, therefore, necessary to apply machine learning in the link adaptation selection process for better optimization since the hybrid link adaptation method achieves lower latency and higher coverage than any other coverage enhancement technique [42].

## 7. Conclusions

In this research work, we implemented, by means of simulation, the application of NB-IoT wireless communication in an industrial indoor setting with the aim of exploring its potential applications. In this study, there were results collected which revealed that the environment and equipment have an impact on wireless communication. Using the simulated throughput results as an example, it can be seen that the maximum effective throughputs for NB-IoT are approximately 21% and 32% less than the peak data rates of 226.7 kbps and 250 kbps for the downlink and uplink, respectively, as proposed in the literature [35]. As a comparison, these results were compared to the theoretical results based on the same assumptions. From the compared results, it can be seen that the theoretical values are slightly higher.

## Figures and Tables

**Figure 1 sensors-22-09039-f001:**
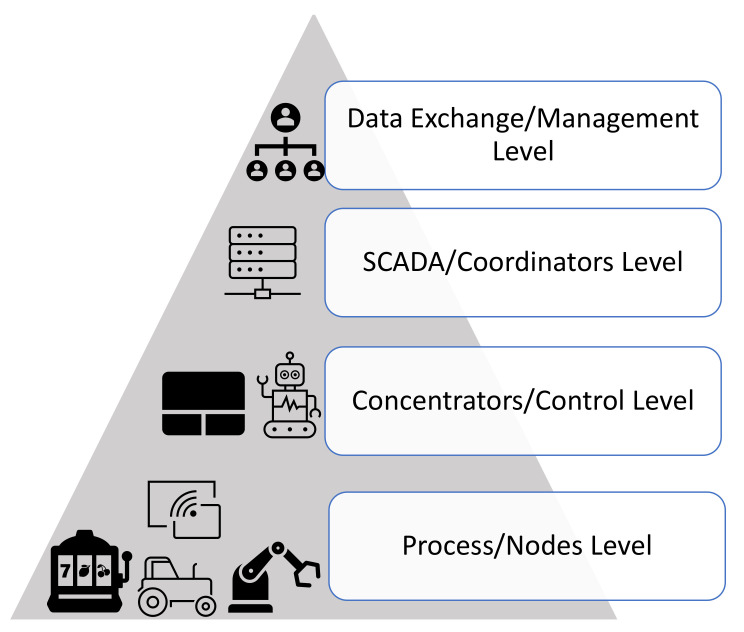
Structured process automation showing four critical levels of data processing.

**Figure 2 sensors-22-09039-f002:**
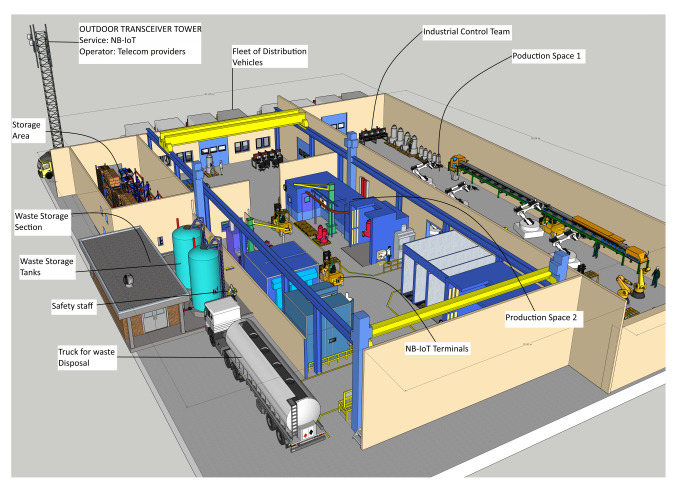
A top overview of the modeled industrial indoor environment. The transceiver tower is positioned outside the structure.

**Figure 3 sensors-22-09039-f003:**
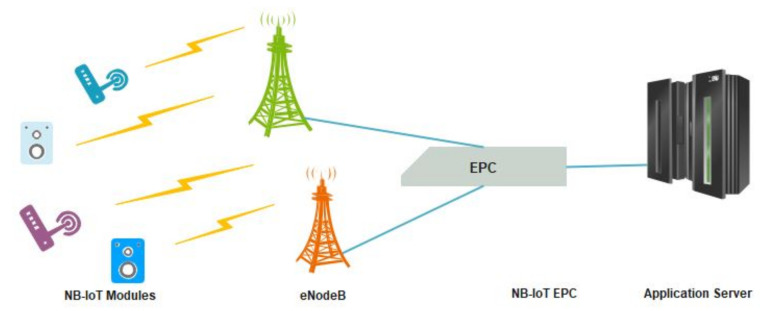
The Infrastructural Architecture of NB-IoT Network.

**Figure 4 sensors-22-09039-f004:**
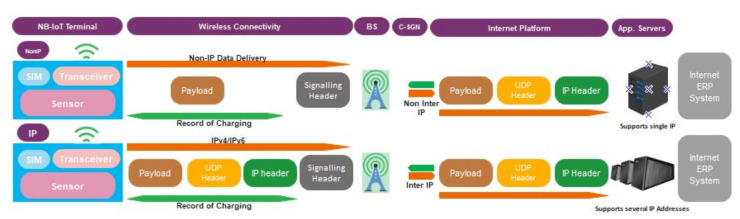
Internal Data Transmission in NB-IoT Network.

**Figure 5 sensors-22-09039-f005:**
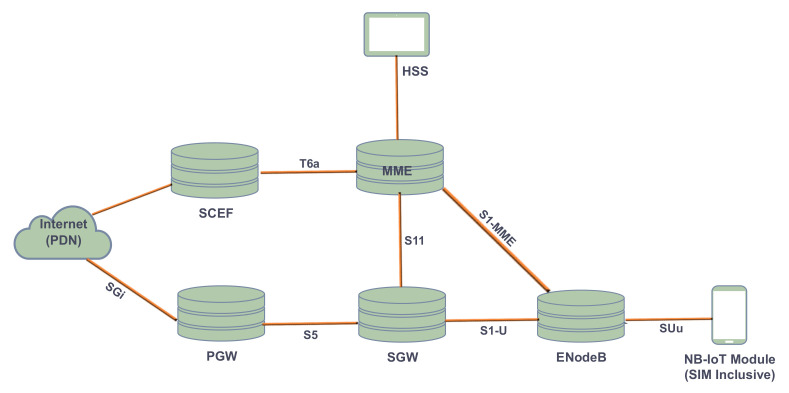
Evolve Packet Core (EPC) Architecture for Data Transmission between NB-IoT Terminals and the Internet using LTE Platform.

**Figure 6 sensors-22-09039-f006:**
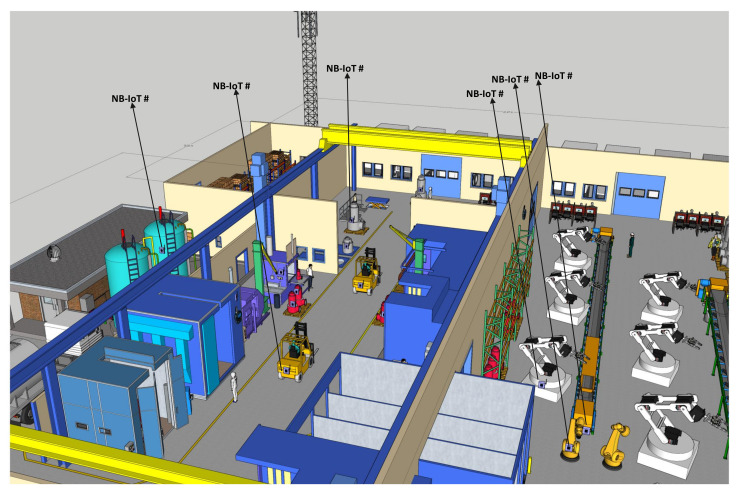
Diagram showing some of the positions of the NB-IoT terminals around the robotic arms, forklifts, and waste tanks.

**Figure 7 sensors-22-09039-f007:**
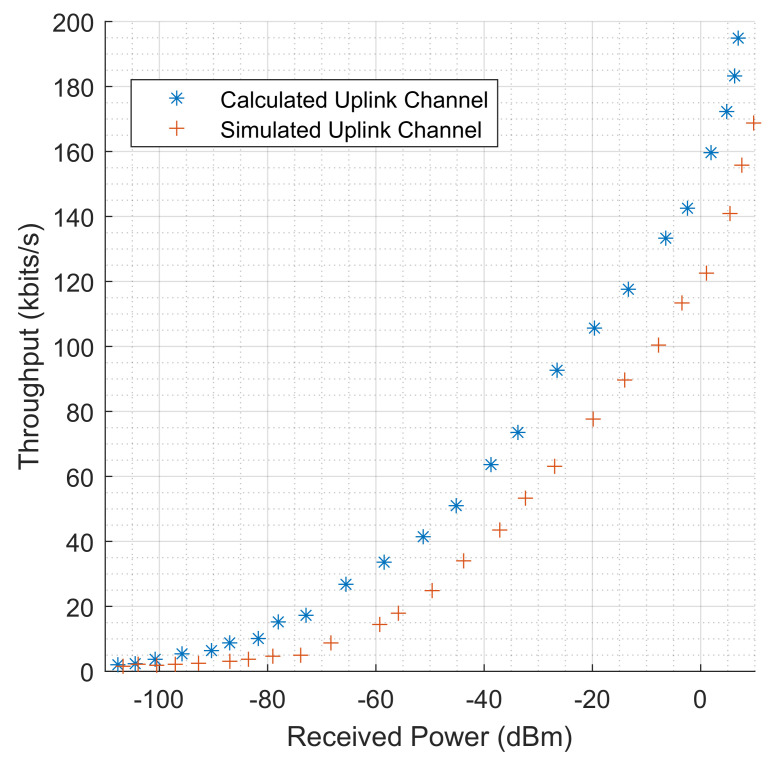
Both simulated and calculated throughput measurements (kbits/s) for NB-IoT uplink channels which are based on the received power (dBm).

**Figure 8 sensors-22-09039-f008:**
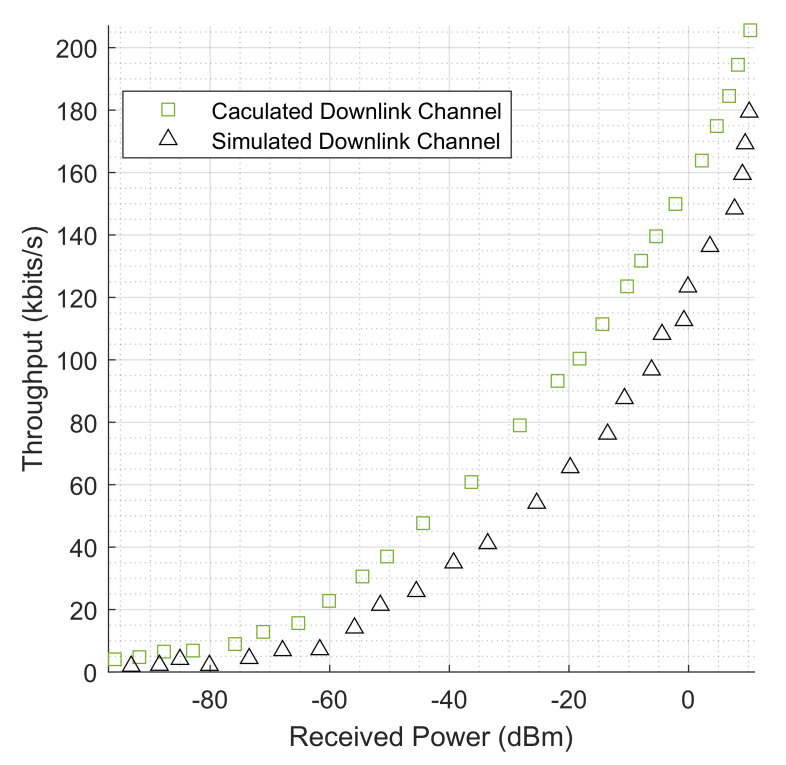
Both simulated and calculated throughput measurements (kbits/s) for NB-IoT downlink channels which are based on the received power (dBm).

**Figure 9 sensors-22-09039-f009:**
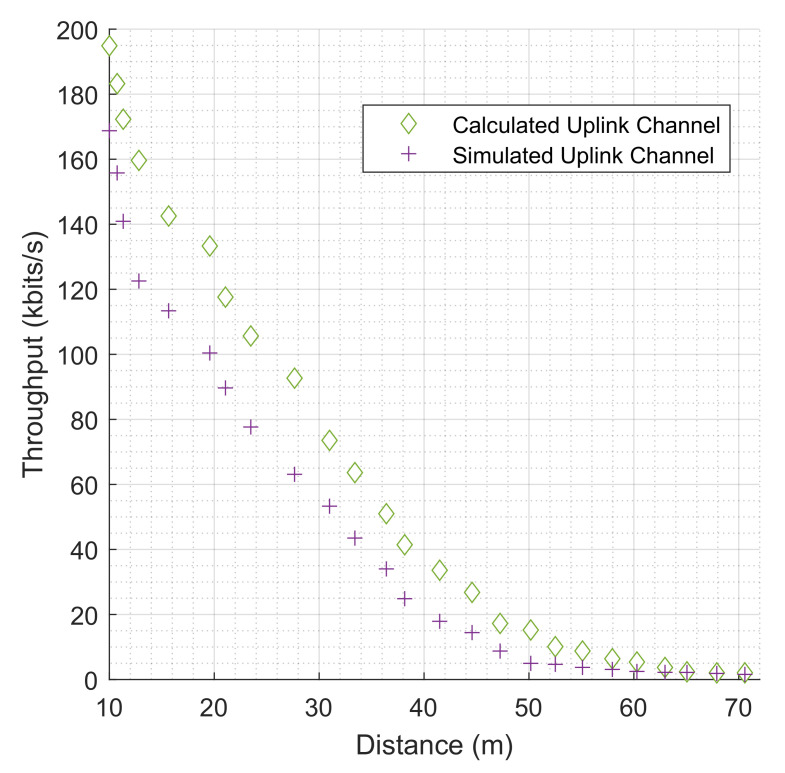
The evaluation of throughput (kbits/s) for both simulated and calculated NB-IoT uplink channels based on distances from the transceiver.

**Figure 10 sensors-22-09039-f010:**
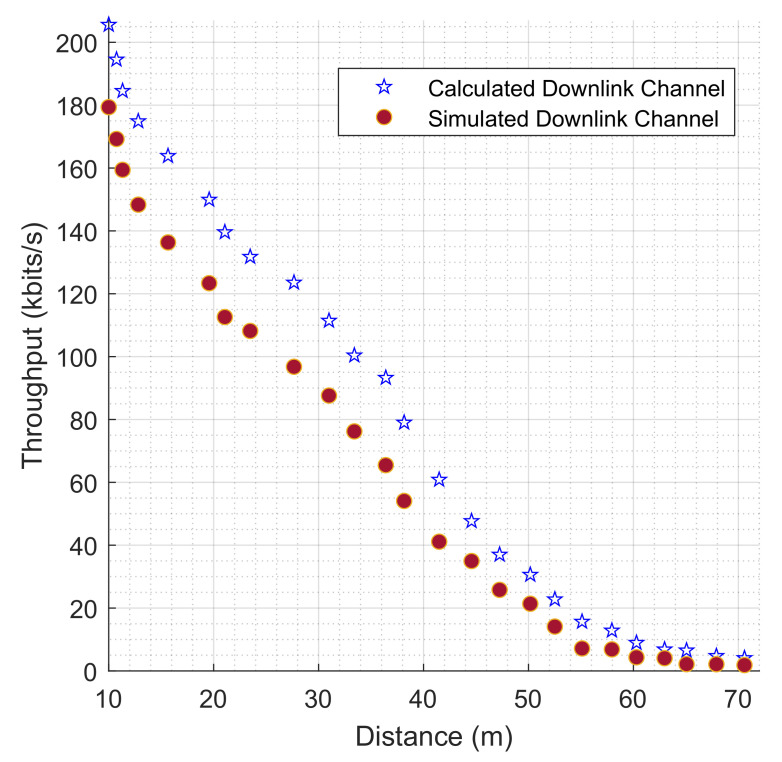
The evaluation of throughput (kbits/s) for both simulated and calculated NB-IoT downlink channels based on distances from the transceiver.

**Figure 11 sensors-22-09039-f011:**
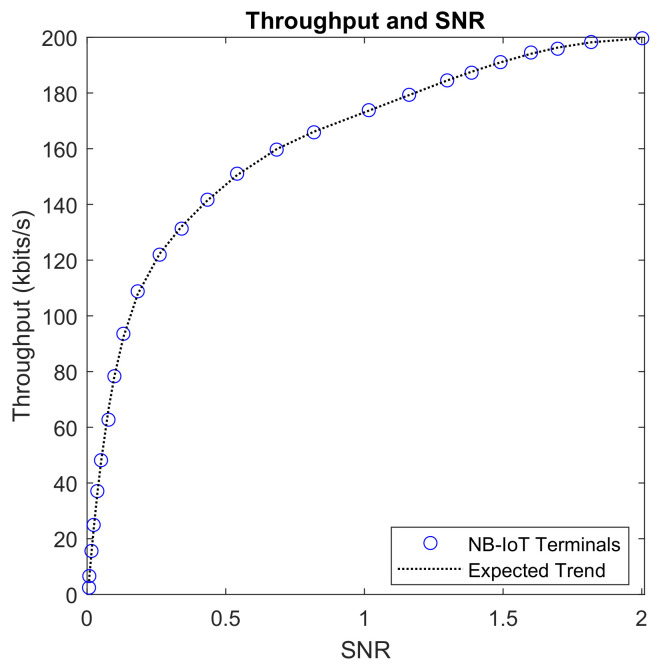
The Effect of SNR on Throughput.

**Figure 12 sensors-22-09039-f012:**
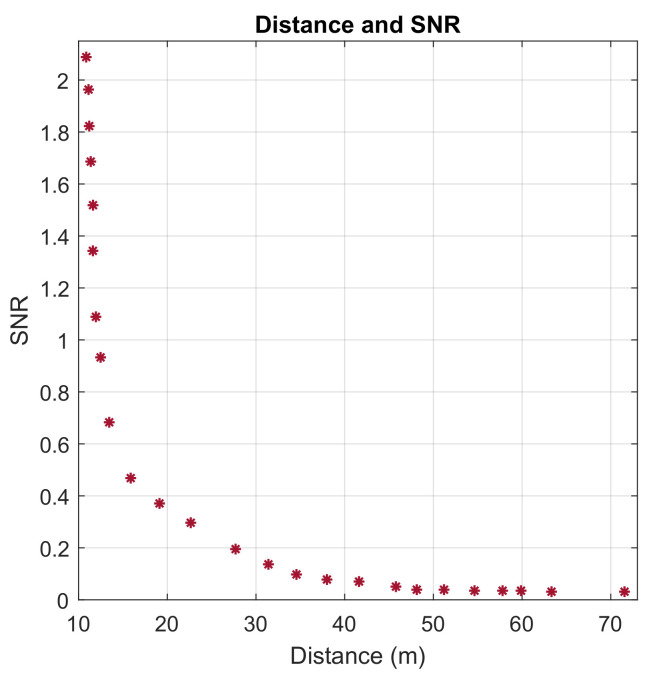
The Relationship between SNR and Distance (m).

**Figure 13 sensors-22-09039-f013:**
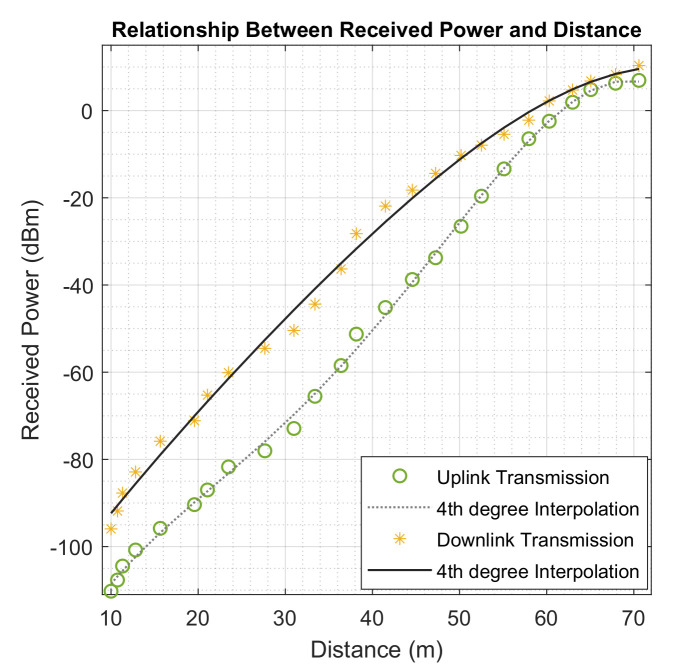
Uplink and Downlink Received Power and Dependency on NB-IoT Distances from the Transceiver.

**Table 1 sensors-22-09039-t001:** WI parameter settings used for simulation of the modeled scenario.

Property Type	Setting
Transceiver antenna	Omnidirectional
Antenna gain	3.5
Antenna waveform	Sinusoidal
NB-IoT max. Tx power	14 dBm
Bandwidth	180 kHz
Deployment mode	In-band
Frequency	900 MHz
Modulation	QPSK
Max. propagation paths	25
Propagation model	X3D
Number of reflections	6
Number of transmissions	4
Number of diffractions	1
Receiver threshold (dBm)	−250
Ray tracing acceleration	Octree
Ray spacing	0.2500
Polarization	Vertical
Tx/Rx coordinate system	Cartesian

## Data Availability

Not applicable.

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
