# Peer review of "Towards the Digital Twin (DT) of Narrow-Band Internet of Things (NBIoT) Wireless Communication in Industrial Indoor Environment"

_sensors, 2022, doi:10.3390/s22239039_

Round 1

Reviewer 1 Report

Please check the attached pdf file.

Author Response

For responses, please see the file attached

Reviewer 2 Report

Section 2 presents works carried out with NB-IoT in different environments. What does the new environment provide that is different from what has already been published?

There is talk of industrial environments and finally a very specific one has been chosen. However, a car factory is not the same as a logistics center or a chemical plant. It should be explained how to extrapolate this study to other types of facilities.

Apparently the characteristics of the plant have been chosen arbitrarily. Why has that plant been used?

It would be desirable to better highlight what advantages NB-IoT brings over other data and information transmission methods for the industrial environment in particular. It would help to make the interest of the work clearer.

In lines 289 and 290 a blank space is required between number and units.

Author Response

For our responses, please see the file attached

Reviewer 3 Report

This paper presented a study of the behavior of NB-IoT wireless communication in an industrial indoor environment. The paper looks well-organized and well-written, but I have some comments and suggestions as follows:

- I suggest moving up the paper's contributions and the paper organization from section 2 to section 1. 

- I did see any related work discussing the digital twin of NBIoT. Please provide several studies. 

- The authors should provide the system model(indoor environment structure) used by WI software for more understanding.  

- On page 10, line 429, "These equipment are modeled using a third-party software and imported into WI using the Stereolithography (STL) file format." What is third-party software?

- On page 11, line 440 "To start with, a bandwidth of 180 kHz was set and deployed in the in-band mode as specified by 3GPP" Please provide 3GPP TR/TS number.

- Figure 6 should move from page 12 to page 13. Also, please check the figure numbering on page 13, line 529 "Figure ??" and enlarge all figures.

- The authors only focused on the throughput. I suggest considering also latency/delay which is the most important metric for NBIoT. 

Author Response

(The authors gave the same response as above.)

Round 2

Reviewer 1 Report

The modified version is okay.

Author Response

N/A

Reviewer 3 Report

Thanks for addressing all my concerns, but the last response to my comments regarding the latency was not convincing. 5G offers three fundamental performance enhancements:

  • Enhanced mobile broadband (eMBB) will deliver ongoing improvements to mobile data connectivity, characterized by increased throughput and improved coverage and capacity. It will also enable fixed wireless access (FWA) services which can compete with fixed broadband.
  • Ultra-reliable and low-latency communications (URLLC) will enable the mass adoption of internet of things (IoT) services across many verticals. This results in reliable indoor coverage and the ability to support a very high density of devices.
  • Massive machine-type communications (mMTC) will offer connectivity for mission-critical applications that require very low latency and high reliability and security.

The eMBB component is more focused on throughput while URLLC is the most significant paramter for IoT. Hence, I suggest the authors include one figure to discuss IoT latecny (if possible). Please refer to the work  "A survey on deep learning for ultra-reliable and low-latency communications challenges on 6G wireless systems" and include it in section 1 or 2. 
